# Age and Hair Cortisol Levels as Predictors of SARS-CoV-2 Infection

**DOI:** 10.3390/ijerph21091166

**Published:** 2024-09-02

**Authors:** Nancy Fiedler, Pamela Ohman-Strickland, Jialin Doris Shen, Kathleen Black, Daniel B. Horton, Reynold Panettieri, Martin J. Blaser, Jeffrey Carson, Kestutis Bendinskas, Hana Cheng, Emily S. Barrett

**Affiliations:** 1Environmental and Occupational Health Sciences Institute, Rutgers School of Public Health, Piscataway, NJ 08854, USA; ohmanpa@sph.rutgers.edu (P.O.-S.); kgb3@eohsi.rutgers.edu (K.B.); esb104@eohsi.rutgers.edu (E.S.B.); 2Pfizer Inc., 400 Crossing Blvd, Bridgewater, NJ 08807, USA; doris.shen@pfizer.com; 3Department of Pediatrics, Rutgers Robert Wood Johnson Medical School, Institute for Health, Healthcare Policy, and Aging Research, Rutgers Biomedical and Health Sciences, 112 Paterson St., New Brunswick, NJ 08901, USA; daniel.horton@rutgers.edu; 4Rutgers Institute for Translational Medicine and Science, New Brunswick, NJ 08901, USA; rp856@rbhs.rutgers.edu; 5Robert Wood Johnson Medical School, Center for Advanced Biotechnology and Medicine, Rutgers University, 679 Hoes Lane West, Piscataway, NJ 08854, USA; martin.blaser@cabm.rutgers.edu; 6Department of Medicine, Rutgers Robert Wood Johnson Medical School, 1 Robert Wood Johnson Place, New Brunswick, NJ 08901, USA; carson@rwjms.rutgers.edu; 7Chemistry Department, State University of New York at Oswego, 30 Centennial Drive, Oswego, NY 13126, USA; kestutis.bendinskas@oswego.edu (K.B.); hanacheng2028@u.northwestern.edu (H.C.)

**Keywords:** SARS-CoV-2, hair cortisol, chronic stress, COVID-19, health care

## Abstract

Chronic psychosocial stress is known to adversely impact immune function. During the SARS-CoV-2 pandemic, occupational stress among workers in healthcare was at an unprecedented level due to risks of infection and work demands. We performed a nested case–control study to investigate the associations between chronic stress and the risks of contracting SARS-CoV-2. We collected 3 cm of hair from employees at an academic medical center who tested positive for SARS-CoV-2 (N = 49) and controls who tested negative (N = 49), matched for age, race, and sex. The diagnosis of SARS-CoV-2 was based on polymerase chain reaction or antibody tests. As a proxy for chronic stress, we segmented hair into 1 cm sections each representing one month and measured cortisol levels using a cortisol enzyme-linked immunosorbent assay. For cases, we used cortisol concentrations measured in hair segments from the month prior to a positive SARS-CoV-2 test, and for controls, we used time-matched hair segments. We fitted conditional logistic regression models adjusted for sex, age, race, body mass index, and healthcare worker status, and stratified models by older vs. younger age (cutoff = 41 years). African Americans had higher hair cortisol levels relative to participants of other races and ethnicities. In adjusted models, higher hair cortisol concentrations were associated with an increased odds of infection with SARS-CoV-2 (OR = 1.84; CI: 1.10–3.07) among older, but not younger, participants. The results suggest that psychosocial stress may be a risk factor for SARS-CoV-2 infection; stress management may be an important part of a comprehensive approach to protect against SARS-CoV-2 infection.

## 1. Introduction

Social determinants of health are recognized as significant factors in vulnerability to a broad range of exogenous environmental exposures such as toxic chemicals, viruses, traumatic events (e.g., earthquakes), and physical challenges [1,2,3,4,5,6,7,8,9,10]. Numerous studies document that poverty, minority status, and low education increase vulnerability to a range of adverse health outcomes because of the biological alterations associated with these conditions [11]. That is, when individuals are chronically exposed to adverse conditions in their home, neighborhood, and work environments, these conditions induce chronic activation of the hypothalamic–pituitary–adrenal axis (HPA axis). For example, Neppl et al. (2024) [12] reported that economic pressures experienced in adolescence were significantly positively correlated with hair cortisol levels measured among those adolescents as adults. The HPA axis is involved in the release of cortisol, epinephrine, and norepinephrine in response to stressors. These hormones have cascading effects on multiple organ systems, including immune function [5,13,14]. For example, numerous studies document that chronic stress increases vulnerability to respiratory infections such as the common cold and flu [6]. Other investigators have also hypothesized HPA axis dysregulation as a mechanism for susceptibility to SARS-CoV-2 infection [15]. During the early stages of the COVID-19 pandemic, employees in medical communities experienced numerous chronic stressors to include long working hours, care for critically ill patients, and a lack of protective equipment, increasing the risk of exposure to the virus [16,17].

Several studies suggest that salivary cortisol, as a biomarker of stress, may heighten susceptibility to the common cold [18,19,20]. For example, in three well-controlled studies involving nasal instillation of the rhinovirus, higher salivary cortisol levels in the 3 days prior to instillation were associated with a greater likelihood of infection and a significantly longer duration of viral shedding [19]. In addition, our controlled exposure studies document the relative importance of an acute psychosocial stressor in reported health effects from exposures to volatile organic mixtures [21]. Cortisol, as a marker of HPA function, can be measured in several matrices including blood, urine, saliva, hair, and nails. Gold standard approaches for measuring cortisol include assessing diurnal variations in multiple samples collected at specific times across the day, as well as cortisol reactivity in response to an acute stressor. Given the high burden of those sampling approaches, a more practical alternative is the collection of hair samples, which can be used to estimate cortisol production over longer time periods [16,17]. Elevated hair cortisol levels have been seen among earthquake survivors [2] and have been associated with stressful life events [4,22,23,24]. Test–retest serial measures of hair cortisol (HCC) levels are highly reliable (e.g., r = 0.84), with moderately high intra-class correlations [25] and correspondence with integrated salivary cortisol measurements [26,27]. Although hair treatments could affect cortisol measurements [28], evidence suggests that hair washing and coloring do not alter HCC measurements [29].

Our ongoing study to evaluate predictors of susceptibility to SARS-CoV-2 included healthcare and non-healthcare faculty and staff within a northeastern USA university community who were enrolled near the start of the U.S. COVID-19 pandemic between March and April 2020 [30,31]. Prior observations based on this cohort indicated that the risk of SARS-CoV-2 infection early in the pandemic was higher among healthcare workers (prevalence: 7%) than non-healthcare workers (prevalence: 0.4%), with nurses and those who worked with more patients with SARS-CoV-2 at greater risk [32]. A subsequent evaluation of a larger group of hospital employees revealed that healthcare workers at the greatest risk of SARS-CoV-2 infection were support staff and underrepresented minority (URM) groups with and without patient care responsibilities [32]. The investigators hypothesized several factors that may have contributed to higher rates among support staff relative to physicians and nurses. For example, support staff may have had less access to personal protective equipment and more of a likelihood of community-based exposures. Another potential explanatory factor could be that lower paid workers from minority communities may have also experienced more stress in their communities and homes during the pandemic. Although the literature clearly documents that chronic stress increases susceptibility to respiratory illness, no study to date has investigated whether chronic stress also increases susceptibility to SARS-CoV-2. We hypothesized that higher hair cortisol levels, as a biomarker of chronic stress, are associated with an increased risk of SARS-CoV-2 infection while controlling for sociodemographic factors. We explored this association in groups stratified by race, age, BMI, and healthcare worker status.

## 2. Materials and Methods

### 2.1. Participants

From the initial cohort of 548 healthcare workers and 283 non-healthcare workers from an academic medical center, 377 participants consented to the collection of hair samples from May to June 2020. Eligibility criteria for the initial cohort included healthcare workers with regular direct patient contact (e.g., ≥20 h/week) and non-healthcare workers without patient contact. Volunteers who were pregnant or breastfeeding, had a previous diagnosis of SARS-CoV-2, or with a newly diagnosed medical condition (past 30 days) or a change in medication were excluded (see [26] for complete recruitment details). We identified from the 377 participants with hair samples, 49 participants who provided hair samples in the month prior to testing positive for SARS-CoV-2 through a polymerase chain reaction (PCR) or antibody (AB+) test [30]. Each case was matched to a negative control within the cohort, defined as having had no evidence of SARS-CoV-2-infection within 3 months of hair sampling (total N = 49). Controls were matched based on the following variables: age (±10 years), self-reported race (White, Asian, Black, or other), sex, and an available hair segment for analysis. The hair segment for cases was the segment representing the hair cortisol level one month prior to the date of SARS-CoV-2 diagnosis. Because the hair cortisol level declines with greater distance from the scalp, we matched the hair segment used for controls with the segment used for cases [33]. All study activities were approved by the Rutgers University Institutional Review Board and all participants signed informed consent forms prior to participation.

### 2.2. HCC Measurement Method

The hair analysis was performed in Dr. Bendinskas’ laboratory, as previously described [34]. Hair was collected by trained study staff at one of the planned cohort visits (at week 4 or 8 following enrollment). At the time of hair collection, participants completed a brief questionnaire to collect data on hair product use. We cut approximately 150 strands of hair at the posterior vertex of the subject’s head as close to the scalp as possible, yielding at least 3 cm of length [35]. Hair grows at approximately 1 cm/month with the 3 cm closest to the scalp reflecting cortisol production over the preceding three months [36,37,38]. Samples were stored at room temperature under dry and dark conditions until they were sent at room temperature to the Bendinskas laboratory at SUNY-Oswego lab for analysis [3,39]. The three cm of hair closest to the scalp were segmented into three 1 cm portions and washed three times with isopropanol at room temperature for 30 s to remove external contamination. Hair was dried at 70 °C in an oven for at least two hours, weighed using a high-sensitivity analytical balance, and milled using 7 mm stainless steel balls. Cortisol was extracted with methanol overnight at 55 °C, acetone for 5 min at room temperature, and then with methanol overnight at 55 °C once more [40]. Pooled solvent fractions were removed under a nitrogen stream. Samples were dissolved in assay diluent, distributed to avoid a batch effect, and analyzed in duplicate using a cortisol enzyme-linked immunosorbent assay (Arbor Assays kit K003-H5) [41]. We repeated the analysis of 5% of all samples randomly to ensure reproducibility and reran any samples whose replicates exceeded 5% relative standard deviation (RSD) or required dilutions. HCC levels (in pg/mg) and relative standard deviation for each 1 cm segment in each sample were reported; thus, three cortisol values were obtained for each participant, one corresponding to each cm of growth (or month of time). For the current analysis, we focused the analysis on the HCC levels in a segment 1 month prior to infection for cases. We chose the same, time-matched segment for each control. The lower limit of cortisol detection was 74 pg/mL in an extract [42] or about 1.2 pg/mg in hair; our intraplate RSD was 2.1%, and the interplate RSD was 7.8%. The median HCC level in the 98 hair samples was 4.9 pg/mg. Linearity (R^2^ = 0.994 in the range of 2.5 to 20 mg of hair), parallelism, and extraction reproducibility (8% variability) experiments were completed to validate the procedures.

### 2.3. Statistics

We summarized age, sex, ethnicity, BMI, and occupation (healthcare worker—yes or no) using frequencies (percentages) for cases and controls. We described the distributions of HCC levels using means, standard deviations, and percentiles, stratified by age, race, sex, and body mass index. These stratifications were chosen based on the previous literature documenting differences in hair cortisol levels [33]. To assess the association between HCC levels and testing positive for SARS-CoV-2 within one month prior to infection, we used a conditional logistic regression model with testing positive for SARS-CoV-2 as the outcome and HCC as the predictor, conditioning on case–control pairs. We also examined whether certain factors modified the effects of cortisol by stratifying conditional logistic regression models by sex (male/female), age (based on a median split of <41/>41), ethnicity (White/non-White), BMI (overweight vs. normal weight), or healthcare worker status (healthcare/non-healthcare). We examined interactions between cortisol levels and the potential effect modifier in unadjusted and adjusted conditional logistic regression models. In sensitivity analyses for overall HCC levels and when examining modification effects, we included a continuous value for BMI and healthcare worker status as covariates to examine changes in effect sizes due to confounding.

## 3. Results

Participants were predominantly female (78%), Caucasian (73%), and with a median age of 41 years. Over 50% of participants were overweight or obese. An approximately equal number of healthcare and non-healthcare workers were included in this sub-study. SARS-CoV-2 cases and controls were matched for age, race/ethnicity, and sex (Table 1). Table 2 shows HCC values for cases and controls stratified by age (median split), race (Asian, Black, White, or other), healthcare worker status (yes/no), and BMI (normal weight vs. overweight/obese).

In bivariate analyses, HCC levels were similar in cases compared to controls (Table 2). Additionally, HCC levels did not differ significantly by sex, BMI, or healthcare worker status within groups or across cases and controls. We noted age-related differences in HCC levels. Specifically, older (>41 years) SARS-COV-2 cases had higher cortisol concentrations than controls and younger (<41 years) cases. We also observed race-related differences in HCC levels. Cases who identified as Black had the highest concentration of cortisol among groups, though the sample size was small (N = 4). Based on the observation of higher cortisol concentrations among Blacks in our case–control sample, we evaluated HCC levels among all Blacks (N = 11) within our entire cohort of healthcare and non-healthcare workers and observed higher HCC levels in Blacks relative to Whites (N = 72) for each hair segment 1 (A) to 3 (C) cm from the scalp (mean (SD): segment A: Black HCC = 10.37 (4.7); White HCC = 6.42 (5.11); segment B: Black HCC = 11.81 (7.99); White HCC = 6.42 (9.64); segment C: Black HCC = 14.55 (14.74); White HCC = 6.59 (13.15).

In unadjusted logistic regression models, we observed no increased risk overall of testing positive for SARS-CoV-2 with increased concentrations of HCC. However, among older participants (>41 years of age), the HCC level was associated with 13-fold increased odds of testing positive for SARS-CoV-2 for a one interquartile range (25th–75th percentiles) of the HCC level (OR: 13.64; 1.45, 128.10) (Table 3). The age by HCC interaction was significant (*p* = 0.0096) (Table 3).

Associations between HCC levels and the SARS-CoV-2 infection status remained null after adjustment for the healthcare worker status and body mass index (OR: 1.03; 95% CI: 0.93, 1.13) (Table 4). Results from models stratified by sex, healthcare worker status, or body mass index were similarly null. As in the unadjusted models, when we stratified by age (at a cutoff of median age 41) after adjustment for the healthcare worker status and body mass index, among the older participants, a 1 pg/mg increase in the HCC level was associated with 84% increased odds of SARS-CoV-2 test positivity (OR: 1.84; 95% CI: 1.10, 3.07). Equivalently, a one interquartile range increase (IQR = 3.29) in HCC levels increased the odds of a positive SARS-CoV-2 test for participants > 41 years by 13.64 times the odds before increased HCC. Finally, in models stratified by race (White vs. non-White), among non-White participants, we observed a trend between HCC levels and the odds of SARS-CoV-2 infection; however, the sample size of non-White participants was small (N = 26). The results of sensitivity models, in which cortisol values were log-transformed, were unchanged (Appendix A).

## 4. Discussion

In this nested case–control study, hair cortisol levels were higher in the month prior to the positive SARS-CoV-2 test in cases compared to time-matched controls. Among all participants, we did not observe an increased odds of SARS-CoV-2 with higher hair cortisol levels. However, in older participants (>41 years), hair cortisol levels were significantly higher in those who subsequently tested positive for SARS-CoV-2. These results suggest that among older (but not younger) adults, higher cortisol productivity, indicative of HPA activity, may be associated with increased vulnerability to SARS-CoV-2 infection.

Adrenal changes that naturally occur with aging, particularly after age 40, result in higher mean cortisol levels and, ultimately, an alteration of the negative feedback loop that dampens the immune response [43,44,45]. Cortisol responses to stressors also change with aging. For example, with age, there may be increased cortisol production in response to either pharmacologic or psychologic challenge, and the return to baseline levels may be slower after the cessation of the stressor or a pharmacologic suppression test [46,47]. Most of the literature on aging and cortisol is based on either salivary or plasma/serum cortisol levels; limited evidence indicates that age also predicts a higher HCC level [48]. These changes in the HPA axis with age may contribute to declining immune function and greater susceptibility to infectious diseases such as SARS-CoV-2 and its consequences. Our finding of an association between HCC levels and SARS-CoV-2 infection for subjects over age 41 years is consistent with what is known about HPA axis alterations with age [33]. However, further work is needed to better understand fully the impact of aging on HCC.

Previous studies document the substantial stress and mental health symptoms experienced by healthcare workers during the SARS-CoV-2 pandemic [16,49,50,51]. Relatively few studies, however, document physiologic markers of stress, such as HCC levels, among healthcare workers, and the results have been mixed regarding the relationship between HCC levels and perceived stress/burnout [51,52,53]. In some studies, a higher cortisol level was associated with increased reports of burnout [51], while others found no associations with self-reported psychological symptoms such as perceived stress [33,53]. Studies evaluating stress, mental health symptoms, and burnout associated these outcomes with HCC levels but did not use HCC levels to predict SARS-CoV-2 infection (e.g., [51,52]). For example, Ibar et al. reported that depersonalization, as one aspect of burnout, mediated the relationship between perceived stress and higher HCC levels. Our healthcare workers, however, did not have higher mean hair cortisol levels relative to non-healthcare workers and were not at an increased risk of contracting SARS-CoV-2 infection.

In a series of controlled viral challenge studies, Cohen and colleagues explored the psychosocial factors associated with increased susceptibility to respiratory illnesses among healthy subjects (e.g., [54]). This body of work is directly relevant to the current study in its emphasis on factors that increase susceptibility to respiratory illnesses, including the common cold and flu. For example, Cohen et al. (1998) [55] reported that chronic stressors lasting more than one month, such as work stress (underemployment), increased the odds of developing a cold from an intentional and controlled instillation of a rhinovirus. In a subsequent study, subjects experiencing a sleep duration <7 h compared to those sleeping ≥8 h and a sleep efficiency <92% compared to those with a sleep efficiency >98% were more likely to develop a cold following viral challenge [56].

Acute stress increases cortisol levels, which in turn reduces the production of inflammatory cytokines as protection from the inflammatory response [3]. However, when stress becomes chronic, the biological mechanisms that account for the effects of cortisol on inflammation and ultimately chronic illness remain unresolved. In a meta-analysis, Miller et al. (2007) [57] reported that chronic psychological stress dysregulates cortisol, leading to persistently higher levels of cortisol. Although cortisol, as an anti-inflammatory, is expected to suppress peripheral inflammatory markers, several animal and human studies observed that a chronic elevation of cortisol levels is associated with systemic inflammation, as indicated by elevations in the levels of inflammatory markers such IL-6 (Interleukin-6) and CRP (C-reactive protein). Thus, chronic stress has the opposite effect of acute stress that may be attributable to glucocorticoid resistance or glucocorticoid receptor alterations resulting in resistance to glucocorticoids in monocytes [16,58,59,60,61]. Growing evidence supports the development of glucocorticoid resistance over a one-year period among individuals experiencing significant and ongoing chronic stress related to caregiving of a family member with a chronic illness [60,62,63]. For example, Miller et al. (2002) [62] observed that parents of children treated for cancer (chronic stress) did not show the expected dampened inflammatory response to dexamethasone compared to parents of healthy children whose blood samples showed suppressive responses to dexamethasone, as expected. In a follow-up study, Cohen et al. (2012) [64] documented lower immune cell sensitivity to cortisol among subjects with stressful interpersonal experiences and an increased risk of developing a cold following exposure to a rhinovirus. Cohen et al. (2021) [5] summarized these investigations and concluded that chronic stress, possibly of the type experienced by healthcare workers, can dysregulate immune modulation by cortisol, thereby increasing susceptibility to respiratory infections. In a meta-analysis of hair cortisol studies, Stalder et al. (2017) [33] reported cortisol hypersecretion with chronic stress, with the highest increase when chronic stress is continuing rather than in the past. This persistent chronic stress may dysregulate the HPA axis, resulting in susceptibility to SARS-CoV-2 infection, as hypothesized by Cohen (2021) [15]. We did not observe an increased risk of SARS-CoV-2 based solely on classification as a healthcare worker, even though the literature documents the significant psychological stress and likely poor sleep related to work demands during the pandemic [65]. However, the working conditions experienced by the healthcare community are consistent with the increased susceptibility to respiratory illness associated with chronic stress [5,58,66].

Although based on a small sample, we observed the highest HCC levels among Black participants and an increased risk of SARS-CoV-2 infection. This finding is consistent with previous studies comparing HCC levels between Whites and Blacks, with evidence of an association between higher HCC levels and perceived discrimination for Blacks that may be mediated by psychological symptoms such as anxiety [67,68,69,70]. Moreover, several studies report a dysregulated and/or dampened cortisol response because of perceived discrimination among Blacks [39,69,71]. Our results are suggestive and provide motivation to explore how discrimination affects HPA axis dysregulation and increases the risk of respiratory infections.

The findings of our study offer intriguing results that suggest directions for future research into the risks of SARS-CoV-2 and other respiratory illnesses. However, our study had several limitations, including the small sample size, particularly for URM participants, and variability in the timing of the HCC measurement relative to the diagnosis of SARS-CoV-2 infection. In addition, the recruitment of participants from an academic medical setting, who had not already been diagnosed with SARS-CoV-2 before obtaining a hair sample, may have led to a somewhat homogeneous sample, reducing the power and generalizability. Our results add to the small literature suggesting that stress, and particularly HPA axis activity, may confer vulnerability to infectious agents such as SARS-CoV-2 among older adults. Our findings complement prior research on the effects of age, race, and stress on the HPA axis, immune function, and susceptibility to infections and point to the need to consider these factors as potential risk factors, particularly in future infectious disease pandemics.

## 5. Conclusions

Several studies document the stress experienced by healthcare communities due to the COVID-19 pandemic. To our knowledge, ours is the only study to use hair cortisol levels, as a biomarker of chronic stress, to predict the association of stress with COVID-19. In adjusted models, we observed that older participants (>41 years) had an increased odds of COVID-19 relative to participants ≤ 41 years of age. Moreover, non-White participants tended to have increased odds of developing COVID-19 associated with higher hair cortisol levels. These findings suggest that further studies should consider chronic stress as a predictor of COVID-19 and that preventive measures to reduce stress could also influence the odds of contracting COVID-19.

## Figures and Tables

**Table 1 ijerph-21-01166-t001:** Study demographics.

Characteristics	Cases (N = 49) N (%)	Controls (N = 49) N (%)
Gender		
Female	38 (78)	38 (78)
Male	11 (22)	11 (22)
Age Range (years)		
Mean (SD)	43.9 (13.7)	43.7 (13.2)
20–29	6 (12)	6 (12)
30–39	17 (35)	17 (35)
40–49	10 (20)	10 (20)
50–59	7 (14)	7 (14)
60–79	9(18)	9 (18)
Age (Median)		
≤41 years of age	25 (51)	25 (51)
>41 years of age	24 (49)	24 (49)
Race		
Asian	4 (8)	4 (8)
Black	2 (4)	2 (4)
Other	7 (14)	7 (14)
White	36 (73)	36 (73)
Healthcare Worker		
No	22 (45)	21 (43)
Yes	27 (55)	28 (57)
Body Mass Index		
Mean (SD)	26.7 (4.9)	27.5 (6.8)
Normal	20 (41)	17 (35)
Overweight/Obese ^1^	29 (59)	32 (65)

^1^ Overweight: BMI > 25; Obese: BMI > 30.

**Table 2 ijerph-21-01166-t002:** HCC level (pg/mg) by demographics.

Characteristics	Cases (N = 49)		Controls (N = 49)	
	Mean (SD)	Median (Min, Max)	Mean (SD)	Median (Min, Max)
Gender				
Female	5.9 (5.3)	4.6 (1.4, 29.0)	5.4 (4.4)	4.4 (0.5, 20.6)
Male	5.1 (3.7)	4.0 (2.0, 15.5)	5.3 (3.0)	4.1 (2.2, 11.6)
Age Range				
20–29	7.1 (5.7)	4.5 (3.2, 18.4)	9.3 (8.2)	8.7 (0.5, 20.6)
30–39	4.0 (3.6)	3.5 (1.4, 17.0)	6.2 (4.1)	5.1 (3.0, 19.1)
40–49	5.8 (3.9)	4.6 (1.7, 15.5)	4.0 (1.6)	3.4 (2.4, 7.0)
50–59	9.4 (9.0)	7.6 (2.1, 29.0)	3.6 (1.5)	2.8 (1.9, 6.1)
60–69	5.9 (2.8)	5.5 (2.0, 9.8)	4.8 (1.5)	4.5 (2.5, 7.1)
70–79	3.0 (0.2)	3.0 (2.9, 3.2)	2.2 (0.0)	2.2 (2.1, 2.2)
Age (Median)				
≤41 years of age	4.9 (4.2)	3.9 (1.4, 18.4)	6.9 (5.2)	5.1 (0.5, 20.6)
>41 years of age	6.6 (5.7)	5.4 (1.7, 29.0)	3.9 (1.6)	3.6 (1.9, 7.1)
Race				
Asian	9.4 (6.5)	7.5 (4.2, 18.4)	8.9 (8.0)	5.8 (3.4, 20.6)
Black/AA	18.9 (14.3)	18.9 (8.8, 29.0)	5.4 (1.0)	5.4 (4.7, 6.1)
Other	4.6 (2.5)	3.6 (1.5, 7.8)	3.7 (1.1)	3.7 (1.9, 5.4)
White	4.8 (3.4)	4.0 (1.4, 17.0)	5.3 (4.0)	4.3 (0.5, 19.1)
Healthcare Worker				
No	5.6 (5.6)	4.1 (1.7, 29.0)	5.0 (3.7)	3.9 (2.1, 19.1)
Yes	5.8 (4.5)	4.6 (1.4, 18.4)	5.7 (4.5)	4.5 (0.5, 20.6)
Body Mass Index				
Normal	5.4 (3.9)	4.6 (1.4, 18.4)	4.8 (4.0)	3.9 (0.5, 19.1)
Overweight/Obese	6.0 (5.7)	4.0 (1.5, 29.0)	5.7 (4.3)	4.7 (1.3, 20.6)

**Table 3 ijerph-21-01166-t003:** Unadjusted associations between HCC levels and the SARS-CoV-2 infection status *.

	Odds Ratio OR (CI)	*p*
Overall		
All	1.10 (0.71, 1.70)	0.68
Gender		
Female	1.13 (0.70, 1.81)	0.62
Male	0.92 (0.26, 3.20)	0.89
Age **		
≤41 years	0.51 (0.22, 1.20)	0.12
>41 years	13.64 (1.45, 128.10)	0.022
Race		
Non-White	3.72 (0.39, 35.42)	0.25
White	0.85 (0.48, 1.51)	0.58
Healthcare Worker		
No	1.33 (0.23, 7.87)	0.75
Yes	0.90 (0.29, 2.81)	0.86
Body Mass Index		
Normal	0.53 (0.09,3.03)	0.48
Overweight/Obese	1.28 (0.67,2.43)	0.45

* Increasing odds for one interquartile range (25th–75th percentiles). ** The interaction effect between age and HCC levels on testing positive was *p* = 0.0096.

**Table 4 ijerph-21-01166-t004:** Healthcare worker status- and body mass index-adjusted associations between HCC levels and SARS-CoV-2 infection.

	Odds Ratio OR (CI)	*p*
Overall		
HCC	1.03 (0.93, 1.13)	0.61
Healthcare Worker Status (ref. = No)	0.92 (0.42, 1.99)	0.83
Body Mass Index *	0.97 (0.90, 1.04)	0.42
Age—stratified analyses **		
≤41 years		
HCC	0.88 (0.75, 1.05)	0.15
Healthcare Worker Status (ref. = No)	1.10 (0.35, 3.40)	0.87
Body Mass Index *	0.90 (0.77, 1.05)	0.19
>41 years		
HCC	1.84 (1.10, 3.07)	0.021
Healthcare Worker Status (ref. = No)	0.53 (0.12, 2.44)	0.42
Body Mass Index *	1.07 (0.92, 1.25)	0.38

* The BMI for this analysis uses the numeric BMI values as a covariate. ** Interpretation of the odds ratios in this table: when the HCC level increases by 1 pg/mg, the odds of a positive SARS-COV-2 test for participants aged > 41 years are 1.84 times the odds before the increase. This translates into an odds ratio of 13.64 when the HCC level increases by 1 interquartile range (IQR = 3.29).

## Data Availability

The raw data supporting the conclusions of this article will be made available by the authors on request.

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
