# Peer review of "Age and Hair Cortisol Levels as Predictors of SARS-CoV-2 Infection"

_ijerph, 2024, doi:10.3390/ijerph21091166_

Round 1

Reviewer 1 Report

Comments and Suggestions for Authors

The study examined several factors potentially relevant to infection with SARS CoV-2. The study was very well-analyzed and included a novel (for existing literature) measurement of cortisol - hair analysis. My primary feeling is that there is nothing new or surprising in the results reported, only the type of respiratory virus examined. Basically, the authors found what most other studies have reported for some time, i.e., individuals who are highly stressed are more prone to illness, and older adults' immune systems decline with age - of course stressed older adults are the most likely to become infected. A part of this can be noted by the dates of articles cited - the Cohen information goes back to the 90"s - this is NOT new information.

Given that, a couple of things make the article feasible for publication - the use of hair to measure cortisol (much better than self-report, but not blood- or salivary) and the indication that Covid acts like every respiratory virus studied - stress levels increase the likelihood of infection, particularly in those with weakened immune systems (e.g., older adults). Covid is a relatively new virus on the scene, thus it's probably a good idea to note how it does/doesn't act like other viruses (at least in its present state). Stress and Covid infection have been examined multiple ways in the last few years - this is simply a novel assessment approach.

Author Response

Comment 1: The study examined several factors potentially relevant to infection with SARS CoV-2. The study was very well-analyzed and included a novel (for existing literature) measurement of cortisol - hair analysis. My primary feeling is that there is nothing new or surprising in the results reported, only the type of respiratory virus examined. Basically, the authors found what most other studies have reported for some time, i.e., individuals who are highly stressed are more prone to illness, and older adults' immune systems decline with age - of course stressed older adults are the most likely to become infected. A part of this can be noted by the dates of articles cited - the Cohen information goes back to the 90"s - this is NOT new information.

Given that, a couple of things make the article feasible for publication - the use of hair to measure cortisol (much better than self-report, but not blood- or salivary) and the indication that Covid acts like every respiratory virus studied - stress levels increase the likelihood of infection, particularly in those with weakened immune systems (e.g., older adults). Covid is a relatively new virus on the scene, thus it's probably a good idea to note how it does/doesn't act like other viruses (at least in its present state). Stress and Covid infection have been examined multiple ways in the last few years - this is simply a novel assessment approach.

Response: Thank you for your review of our manuscript and pointing out the context for our findings.  We agree that the association between biomarkers of psychosocial stress with risk for respiratory infection is not a new finding and is well documented in the literature starting in the 1990s.  Although several studies document self report of stress among health care workers during the COVID pandemic, our aim was to investigate whether a biomarker of chronic stress, i.e., hair cortisol, would suggest susceptibility for COVID infection instead of reflecting an independent outcome of the pandemic, i.e. elevated stress biomarker.  Recognition of the biological risk that chronic stress poses for infection and the subset of workers at greater risk, can help target prevention strategies for health care organizations.

We added the following sentence to acknowledge the literature on stress and risk of respiratory illness while articulating our aim for the study:

Although the literature clearly documents that chronic stress increases susceptibility to respiratory illness, no study to date has investigated whether chronic stress also increases susceptibility to SARS CoV-2.  (line 95-97; pg 2).

Reviewer 2 Report

Comments and Suggestions for Authors

Comments and suggestions for Authors

I read with interest the manuscript entitled Age and hair cortisol as predictors for SARS CoV-2 infection’. The authors analysed the adverse impact of chronic psychosocial stress on immune function and investigated the associations between chronic stress and risks of contracting SARS COV-2.

I appreciate the work of the authors and their research, even on a small group of participants, to highlight the physiological (immune) bases through which stress can increase susceptibility to infectious diseases.

Abstract:

The authors should shorten the information regarding the laboratory tests (e.g. to eliminate the explanations in parentheses); all this information is presented in detail in the Material and Method section.

Introduction:

Line 50: ‘Cohen, 2021’ the author's name appears instead of the corresponding number.  Authors should verify the numbers and the reference list (for example line 265 Cohen (2021) [11]’, but in the reference list, line 349, is written ‘11. Jya et al.’).

Line 56: authors should rephrase ‘… stress, defined by measurement of salivary cortisol…’, because it can lead to interpretations; this is not the definition of stress.

Lines 68-70: the explanations related to hair growth would be more suitable for the Material and Method section.

Lines 77-95: The authors should state more clearly and systematized the purpose and hypotheses of the study.

When using any abbreviated term for the first time, the authors should write what the abbreviation means (e.g. line 72 ‘HCC’).

Material and Method:

Authors should provide more information regarding the participants (the method of sampling, inclusion/ exclusion criteria, etc.). Also, is not clear what means ‘the initial cohort’; in the abstract is mentioned about employees at an academic medical center’, but this information is not written in the manuscript.

Results:

The authors should also detail in the text some information that appears mentioned only in the footer of tables no 3 and no 4

Discussions:

The authors should better emphasize the limitations of the study and explain what ‘other factors’ related to their research may have contributed to these limitations.

References are obsolete, only 35% (20 out of a total of 58) are recent publications (within the last 5 years). Authors should improve this aspect.

Date: 12.08.2024

Author Response

I read with interest the manuscript entitled Age and hair cortisol as predictors for SARS CoV-2 infection’. The authors analysed the adverse impact of chronic psychosocial stress on immune function and investigated the associations between chronic stress and risks of contracting SARS COV-2.

I appreciate the work of the authors and their research, even on a small group of participants, to highlight the physiological (immune) bases through which stress can increase susceptibility to infectious diseases.

Response:  Thank you for your review of our manuscript and for your comments to improve the content,

Abstract:

The authors should shorten the information regarding the laboratory tests (e.g. to eliminate the explanations in parentheses); all this information is presented in detail in the Material and Method section.

Response:  Information clarifying laboratory procedures has been deleted from the abstract with strike through on line 25/26; pg 1.

Introduction:

Line 50: ‘Cohen, 2021’ the author's name appears instead of the corresponding number. Authors should verify the numbers and the reference list (for example line 265 Cohen (2021) [11]’, but in the reference list, line 349, is written ‘11. Jya et al.’).

Response: Thank you for finding this error.  The correct reference number is now inserted on Line 50 in place of Cohen, 2021.  (Line 53; pg 2) Reference 11 was entered in error.

Line 56: authors should rephrase ‘… stress, defined by measurement of salivary cortisol…’, because it can lead to interpretations; this is not the definition of stress.

Response: We agree with this comment and have modified for accuracy as follows: Several studies suggest that salivary cortisol, as a biomarker of stress, ... Line 59/60; pg 2  We also added the reference by Turner-Cobb, 2011 in support of this statement.

Lines 68-70: the explanations related to hair growth would be more suitable for the Material and Method section.

Response: This comment streamlines the introduction.  We deleted the technical information about hair growth on lines 72-74; pg 2.

Lines 77-95: The authors should state more clearly and systematized the purpose and hypotheses of the study.

Response: We agree with this comment and added an acknowledgement of the known effects of chronic stress and susceptibility to respiratory illness followed by the purpose of our study and hypothesis from lines 96 through Line 103; pg 2.

When using any abbreviated term for the first time, the authors should write what the abbreviation means (e.g. line 72 ‘HCC’).

Response: HCC is now defined on first use (line 75; pg2).  Thank you for finding this error.

Material and Method:

Authors should provide more information regarding the participants (the method of sampling, inclusion/ exclusion criteria, etc.). Also, is not clear what means ‘the initial cohort’; in the abstract is mentioned about employees at an academic medical center’, but this information is not written in the manuscript.

Response: We agree that additional information will clarify our participant selection.  We included the location of our sample to match the abstract information and eligibility criteria for the initial cohort to clarify what "initial cohort" means. We also cited our previous publication which contains more specific information about inclusion/exclusion criteria for the initial cohort.  see Lines 107-114; pg 3.

Results:

The authors should also detail in the text some information that appears mentioned only in the footer of tables no 3 and no 4

Response:  We included the footnote information from both tables in the text to clarify the results.  See lines: 200-201 for Table 3 and lines 212-214 for Table 4. pgs 5 and 6.

Discussions:

The authors should better emphasize the limitations of the study and explain what ‘other factors’ related to their research may have contributed to these limitations.

Response: We agree that there are limitations to be acknowledged and included further clarification of the "other factors" to which we alluded.  We reported possible bias in case selection due to the requirement for  an available hair sample one month prior to diagnosis and the small number of URM participants that impeded our ability to investigate subgroup risks.  See lines 312-318; pg 8..

References are obsolete, only 35% (20 out of a total of 58) are recent publications (within the last 5 years). Authors should improve this aspect.

Response: We understand this concern and acknowledge that some references are older.  We reviewed each reference, checked for  recent citations of the references as well as similar citations  and included newer references that were pertinent for our manuscript.  However, we did not eliminate older references such as the literature generated from controlled exposures to respiratory viruses because of the importance of these references for our study.  New references are throughout the manuscript.  In some cases, we did not find more recent references in addition to those we cited.  Note that we included a recent reference for our hair cortisol laboratory procedures (Bendinskas et al., 2024) that includes citations for the same laboratory methods used in our study.  We also kept the original methodologic references for our analytic methods.  The following list is a list of the references by number added to our manuscript. 

7,8,9,10,12, 20, 24, 34, 59, 60,61,63,70

Date: 12.08.2024

Reviewer 3 Report

Comments and Suggestions for Authors

Congratulations on producing a very nice study that adds to a very important question on the role of stress and susceptibility to viral infection. Your findings are very interesting and important, especially in relation to vulnerable members of society who might be exposed to more socio-economic stressors that other parts of the population. I appreciate that there are limitations (which you have documented clearly) but that's the reality of conducting science in real settings, especially during a pandemic.

Minor points

Lines 234 and 241: Missing bracket. E.g., in line 234: ........on (e.g., [.......

Lines 250-251: we might expect that increased HCC as a marker of chronic stress 250 should reduce risk for respiratory illness. Comment: I believe that this comment is incorrect. Increased cortisol during chronic stress should increase the risk of respiratory infection due to prolonged repression of pro-inflammatory cytokines. Do you agree?

Paragraph beginning with Line 249: I feel that the central message is not coming across clearly here (unless I am misunderstanding the content). Acute stress leads to production of cortisol which is anti-inflammatory but short lived and does not lead to increased susceptibility to viral infection. Chronic stress, up to a certain point (three months in this case) produces increased amounts of cortisol on a consistent basis which does lead to an increased susceptibility to viral infection, especially for those aged 41 and older. Prolonged chronic stress can cause adrenal exhaustion and glucocorticoid resistance which leads to immune dysfunction and unregulated pro-inflammatory responses in some cases, which would not make one more susceptible to viral infection. The question for me is: when does dysregulation occur? Does the literature provide an answer here? When does chronic stress switch from making one more susceptible to viral infection to less (because of a hyper-activated immune response as a result of glucocorticoid insensitivity). I appreciate that this is a complicated area but it is important to have a clear message.

Author Response

Congratulations on producing a very nice study that adds to a very important question on the role of stress and susceptibility to viral infection. Your findings are very interesting and important, especially in relation to vulnerable members of society who might be exposed to more socio-economic stressors that other parts of the population. I appreciate that there are limitations (which you have documented clearly) but that's the reality of conducting science in real settings, especially during a pandemic.

Response: Thank you for your review and appreciation for our work.

Minor points

Lines 234 and 241: Missing bracket. E.g., in line 234: ........on (e.g., [.......

Response: Bracket has been fixed and references checked.

Lines 250-251: we might expect that increased HCC as a marker of chronic stress should reduce risk for respiratory illness. Comment: I believe that this comment is incorrect. Increased cortisol during chronic stress should increase the risk of respiratory infection due to prolonged repression of pro-inflammatory cytokines. Do you agree?

Response: Thank you for considering the complexity of this issue and bringing it to our attention.  We agree that if HCC reflects chronic stress that leads to glucocorticoid resistance, then risk of infection should be increased. We corrected our sentence to clarify that acute stress increases cortisol which reduces the inflammatory response. (lines 268-270; pg 7) Your next comment reflects the uncertainty about the point at which this occurs.  We reviewed the literature and cited papers that provide intriguing insights into the issue of susceptibility due to prolonged or chronic stress.

Paragraph beginning with Line 249: I feel that the central message is not coming across clearly here (unless I am misunderstanding the content). Acute stress leads to production of cortisol which is anti-inflammatory but short lived and does not lead to increased susceptibility to viral infection. Chronic stress, up to a certain point (three months in this case) produces increased amounts of cortisol on a consistent basis which does lead to an increased susceptibility to viral infection, especially for those aged 41 and older. Prolonged chronic stress can cause adrenal exhaustion and glucocorticoid resistance which leads to immune dysfunction and unregulated pro-inflammatory responses in some cases, which would not make one more susceptible to viral infection. The question for me is: when does dysregulation occur? Does the literature provide an answer here? When does chronic stress switch from making one more susceptible to viral infection to less (because of a hyper-activated immune response as a result of glucocorticoid insensitivity). I appreciate that this is a complicated area but it is important to have a clear message.

Response: We also agree that prolonged "exposure" to increased cortisol increases susceptibility to viral illness, and that there is uncertainty about the when immune dysregulation occurs.  We reviewed the literature and added further discussion and references for clarification.  However, studies are not uniform and underlying mechanisms remain to be understood.  We hope that our discussion and references on lines 268 through 283  (pg 7)help clarify our points while recognizing uncertainties.  From our review of the literature, it appears that this dysregulation may occur within a year, but exactly how it affects the glucocorticoid receptors and the downstream effects remains unclear.